# The Role of Interdisciplinary Convergence for Mental Health Among Korean Military Servicemen: Focusing on Depression and Salivary Dehydroepiandrosterone-Sulfate

**DOI:** 10.3390/healthcare13040424

**Published:** 2025-02-15

**Authors:** Daeyeon Jang, Wonjung Ryu

**Affiliations:** Department of Social Welfare Counseling and Institute of Social Welfare, Hankyong National University, 327, Jungang-ro, Anseong-si 17579, Gyeonggi-do, Republic of Korea; daeyeon@hknu.ac.kr

**Keywords:** military soldiers, depression, DHEA-S, mental health, interdisciplinary convergence

## Abstract

(1) Background/Objectives: The purpose of this study was to examine the effectiveness of the “Thank You, Sorry, Love” (TSL) program for improving relationships on mental health and hormones of military soldiers in Korea. (2) Methods: We focused on soldiers’ levels of depression and salivary DHEA-S. The effectiveness evaluation of the program used the pretest–posttest control group design, one of the Quasi-experimental design models. The program evaluation used a quasi-experimental design with a pre-test, post-test, and a follow-up for a treatment group and a control group. The sample comprised 40 military soldiers, who were divided into an experimental group (*n* = 20) receiving the TSL intervention and a control group (*n* = 20) without intervention. The study was conducted from June to September 2016, and the effectiveness of the intervention was evaluated through pre-test, post-test, and four-week follow-up tests. (3) Results: Our results revealed that soldiers’ depression and DHEA-S levels significantly differed between the two groups. During the analysis of the program’s effectiveness in which the pre-test was controlled, a statistically significant difference was found between the groups in the post-test (F = 12.666, *p* < 0.01, ES = 0.255) and the follow-up test (F (1, 39) = 13.319, *p* < 0.01, ES = 0.265) in DHEA-S. (4) Conclusions: These findings imply that the TSL program can be an effective intervention for soldiers suffering from depression while helping to produce clinical guidelines that could be made available in the field of military social work in Korea, which are still in their beginning stages.

## 1. Introduction

Depression and stress are extremely common symptoms reported by military soldiers, and their severity may result in an inability to continue serving in the military or even lead to suicide. According to previous studies, 94% of military patients who attempted suicide or were admitted for suicidal ideation exhibited a depressed mood [1], and 80% of soldiers discharged early suffered from mental health issues such as depression and stress. Currently, interventions for depression and stress are primarily offered to military veterans [2], combat veterans [3], and entry-level soldiers [4]. Such programs for the general population of soldiers are limited; however, they are especially important and relevant in Korea because every man is required to serve in the military, regardless of his opinion. Therefore, it is important to determine the effectiveness of intervention programs that can be applied to the general soldier population.

Soldiers experience high levels of stress and depression due to several factors. Generally, soldiers are required to adjust to a completely new living environment and social hierarchy upon entering the military. However, soldiers in their early 20s have lower coping abilities owing to their emotional instability. They are also affected by being in a controlled environment with limited personal freedom and dealing with a lack of supportive resources within the military hierarchy. Therefore, soldiers tend to suffer from chronic stress and depression [5,6]. Previous studies have shown that soldiers experience high levels of stress and depression during the overall training process [7,8]. Because depression and stress among soldiers are primarily caused by the characteristics of the military environment and conflicts with peers, these factors must be considered [9].

According to the stress process model and interpersonal–psychological theory, positive interactions with external resources are crucial for personal mood stabilization. Kruse et al. confirmed that the higher the sense of loneliness, the lower the sense of belonging [10], whereas Limbert et al. found that social support is necessary for soldiers’ psychological stability [11]. As most studies have generally focused on therapeutic methods such as stress coping strategies, anger therapy, and professional counseling to improve mood, the present study aimed to reduce levels of depression and stress among soldiers by facilitating the development of positive relationships with their colleagues. As Korea is a divided country, South Korean men serve in the military through conscription. Therefore, for soldiers, colleagues who live with them 24 h a day are like another family. In this regard, we focused on the “Thank You, Sorry, Love” (TSL) program for improving family relationships.

In general, positive expressions such as “Thank you”, “Sorry”, and “Love” are tools used to express warmth and play a crucial role in influencing individuals’ emotional and physical health [12,13,14]. This is based on the fact that direct expression tends to increase emotional empathy [15] and fosters a positive mindset [16]. Although the TSL program has been shown to have a positive effect on populations under stress (e.g., parents of children with cancer [17] and North Korean female defectors [18]), it has yet to be applied to the general military population.

According to McCann and Pigeau, most soldiers are exposed to communal living, the military’s organizational environment, and a vertical one-sided communication structure [19]. Positive expressions and related activities are expected to improve soldiers’ relationships, communication, and organizational culture. Therefore, the present study examined the effectiveness of the TSL program on military soldiers suffering from depression and stress. Changes in these risk factors were evaluated by comparing the soldiers’ pre- and post-test scores. Moreover, in this study, the effectiveness of the experiment was verified through dehydroepiandrosterone-sulfate (DHEA-S), a representative “steroid hormone” that responds to body stress. Because their DHEA-S levels could be measured to assess their stress levels, those levels complemented the limitations of the self-reported questionnaires. The hormone DHEA-S is primarily utilized as an indicator of stress and aging, and problems such as chronic stress and depression occur when the amount remaining in the body decreases [20,21]. The DHEA-S is inhibited by the hypothalamic–pituitary–adrenal axis in response to stress. Highly stressed individuals have low levels of DHEA-S, which can result in premature aging, immune imbalance, and reduced mental well-being. Finally, previous research has shown that the stress of survival training and related operations decreases DHEA-S levels [22,23]. However, to the best of our knowledge, no study has examined the effectiveness of the TSL program on the stress levels of military soldiers. Therefore, this present study aims to more scientifically verify the effectiveness of the program to prevent stress experienced by soldiers through measuring changes in DHEA-S hormones from the perspective of interdisciplinary convergence after conducting the TSL program in Korean soldiers. This will not only complement the limitations existing in experimental studies on the effectiveness of the counseling field but may also help determine the extent to which the TSL program might be able to help soldiers’ adaptation to the military in Korea or elsewhere.

## 2. Materials and Methods

### 2.1. Research Design

The research design is shown in Table 1. The study was conducted using the pre-test–post-test control group design, one of the Quasi-experimental design models and was conducted with two groups: an experimental group that received the TSL intervention and a control group that did not. The program was implemented over 10 sessions, and for both groups, its effectiveness was verified through a pre-test (before the start of the program), post-test (after the end of the program), and follow-up test (four weeks after the program). To assign soldiers into the experimental and control groups, they were first classified according to their military service (i.e., army or air force) and rank (Korea’s military service includes four levels: private, private first class, corporal, and sergeant). Subsequently, from the four classified groups, the participants were randomly assigned to either the experimental (*n* = 20) or control (*n* = 20) group.

### 2.2. Intervention: “Thank You, Sorry, Love” Program

The TSL program has been shown to be effective in improving adaptation to the culture of an organization and interrelationships, highlighting the limitations of existing programs [24]. As stated earlier, the TSL program is based on the practice of saying “Thank you”, “Sorry”, and “Love” [25]. First, “Thank you” is important for recognizing the value of colleagues and expressing gratitude through language and behaviors. Second, “Sorry” concerns maintaining relationships by apologizing to or forgiving others. Finally, “Love” involves the expression of care and emotion between individuals and the maintenance of positive energy provided by the program [26,27]. Overall, the TSL program provides resilience to individuals, improves their relationships with others, restores intimacy, and promotes positive self-insight [28].

The TSL program has various strengths. First, it can positively transform the military environment by reducing organizational tension, enhancing a collaborative spirit, and securing support resources. More specifically, the positive dynamics of the organization can be improved as individuals interact with others and experience positive stimuli through practical tasks such as “listening, sympathizing, and encouraging one another”, “sharing time”, and “recording memories” [25]. In previous studies on the effectiveness of the TSL program, positive changes were observed in the enhancement of organizational roles [28], commitment of social welfare workers [17], and willingness of self-help project participants [27]. Second, the program can be effective in improving soldiers’ ability to deal with personal crises. Additionally, numerous studies have examined the effectiveness of the TSL program on stress–vulnerable groups such as military personnel, North Korean female refugees [20,26], parents of pediatric cancer patients [17], youth exposed to domestic violence and juvenile delinquents [29]. Furthermore, an important part of this program is exploring the importance of oneself and others through the expressions “Thank You”, “Sorry”, and “Love” [24], which can foster individual perceptions of happiness and improve personal coping abilities.

### 2.3. Participants

The study participants comprised 40 men enlisted in the army and air force. Approval from the Institutional Review Board of Yonsei University (Approval No. 1040917-201606-HRBR-190-02) was obtained in June 2016, and the study was conducted from June to September 2016. For the sampling of study participants, we visited the unit and provided preliminary explanations to the heads of subordinate units and the staff in charge of each military about the purpose and effectiveness of this study. After obtaining approval for the program implementation from the unit commander, a notice for recruitment was posted on the bulletin board within the unit to recruit participants. At this time, the notice of confidentiality under the statistical law was thoroughly made. The soldiers voluntarily participated in the program after being informed (in advance) of its purpose, process, and duration, as well as the risks and benefits of participating. They were also advised about their right to withdraw from the program at any time and about the confidentiality of their personal information.

### 2.4. Sample Size

The G * Power 3 program was used to identify the sample size for this study. Table 2 shows the sample size check results using the G* Power 3 program. Previous studies have examined the effectiveness of existing TSL programs [17,18], along with their effect size (0.774) and power (95%). These figures were referred to for the analyses. The analysis was conducted by setting the α err probe as 0.05, the power (1-β err probe) 95%, the number of groups as two (experimental group and control group), and the number of covariates as one. Finally, a sample size of at least 30 participants was required to test the effectiveness of the intervention.

### 2.5. Measurement

To measure the effectiveness of the TSL program in a sample of soldiers, we used a self-reported questionnaire and biological markers.

#### 2.5.1. Depression

Depression was assessed using the nine-item Patient Health Questionnaire (PHQ-9) based on the DSM-IV diagnostic criteria [30]. The self-report questionnaire to measure depression was conducted three times. The survey took about 20 min to fill out. As soon as the questionnaire was written, it was encoded and stored in a locked state for 5 years, so that only the researcher could read it. Each participant’s degree of depression during the previous week and the PHQ-9 items were rated on a 5-point Likert scale ranging from 1 (never) to 5 (very often). A higher total score indicated a higher level of depression. The Cronbach’s α coefficient for this scale was 0.899.

#### 2.5.2. Salivary Dehydroepiandrosterone-Sulfate (DHEA-S)

We measured the levels of salivary DHEA-S to determine the oxidative stress biomarker [31]. It is the most abundant steroid hormone secreted from the adrenal cortex and plays several functions in the body, including increasing insulin sensitivity, lowering serum lipid levels, and reducing obesity [32]. However, low levels of DHEA-S are found in individuals suffering from depression [21] and chronic stress [33]. Saliva samples (secreted between 9 a.m. and 11 a.m.) were collected during the pre-, post-, and follow-up tests. The participants rinsed their mouths with water, waited for five minutes, and provided 10 mL of saliva in a collection vial using a short plastic straw. This procedure was based on the recommendations of Shirtcliff, Granger, Schwartz, and Curran [34]. The collected saliva was stored in a frozen state and analyzed at the specialized medical research center on the same day. Samples that were encoded through the analysis were immediately discarded by the researcher in charge.

### 2.6. Statistical Methods

High correlations between pre-, post-, and follow-up test scores occur frequently in experimental social science research. A one-way analysis of variance, which ignores this correlation and uses only post-test scores (or a verification method with an independent samples *t*-test), can significantly impair the power of the tests [28,35]. Therefore, an analysis of covariance (ANCOVA) was used to elucidate the effectiveness of the program in the experimental group compared with the control group. Through this approach, the influence of the independent variables can be assessed by controlling the nuisance factor, which can affect both the dependent and independent variables.

To use the ANCOVA for experimental research, two requirements must be satisfied [36]. First, linearity should be found between the pre- and post-test scores and between the pre- and follow-up test scores. To achieve this, each variable was analyzed using a scatter plot, and the pre-, post-, and follow-up test scores for each variable (i.e., depression and DHEA-S) showed a linear distribution. Hence, this assumption was met. Second, equality in the slope should be observed between pre-test scores and group variables. In our study, the interaction between the group variable and pre-test scores was not significant in relation to the post- and follow-up test scores for depression [post-test, *F* = 2.611, *p* > 0.05; follow-up test, *F* = 3.180, *p* > 0.05] and DHEA-S [post-test, *F* = 0.003, *p* > 0.05; follow-up test, *F* = 0.140, *p* > 0.05]. Thus, the requirements for the ANCOVA were met.

## 3. Results

### 3.1. Demographics of the Participants

As shown in Table 3, the participants in the experimental group were between 19 and 28 years of age (*M* = 21.40, *SD* = 1.96) and included privates (*n* = 2), privates first class (*n* = 10), corporals (*n* = 6), and sergeants (*n* = 2). The participants in the control group were between 19 and 22 years of age (*M* = 20.60, *SD* = 0.88) and included privates (*n* = 6), privates first class (*n* = 6), corporals (*n* = 2), and sergeants (*n* = 6).

### 3.2. Changes in Depression Score

Figure 1 shows the change in the depression score. In the pre-test, the experimental group’s depression score was 12.05 (*SD* = 7.2364) and the control group’s score was 4.90 (*SD* = 4.1026). This means that homogeneity between groups was not secured (t = −3.844, *p* < 0.01). Therefore, there is a limitation in analyzing changes between groups of depression in post- and follow-up tests. However, the post- and follow-up test scores indicated that the TSL program significantly reduced depression (see Figure 1). During the analysis of the program’s effectiveness in which the results of the pre-tests were controlled, the differences between the groups in the post-test (F = 15.459, *p* < 0.01, ES = 0.295) and the follow-up test (F = 17.694, *p* < 0.01, ES = 0.841) were statistically significant. Thus, indicating that the TSL program was effective for reducing the depression levels in the experimental group. According to Cohen’s [37] range of effect size, a value of 0.20 or less would be evaluated as small, a range from 0.21 to 0.49 would be evaluated as small-to-medium, a range from 0.50 to 0.79 would be evaluated as medium, and a range of 0.80 or more would be rated as large. Considering Cohen’s range for the effect size coefficient d, the depression score of the post-test would reflect a small-to-medium effect size (d = 0.295) and a large effect size at follow-up (d = 0.841).

### 3.3. Changes in DHEA-S Score

Figure 2 shows the change in the DHEA-S score. The post-test scores indicated that the TSL program significantly improved the DHEA-S levels of the participants (see Figure 2). During the analysis of the program’s effectiveness in which the pre-test was controlled (t = −0.371, *p* = 0.713), a statistically significant difference was found between the groups in the post-test (F = 12.666, *p* < 0.01, ES = 0.255) and the follow-up test (F = 13.319, *p* < 0.01, ES = 0.265). In other words, the TSL program reduced the DHEA-S levels in the experimental group, with a statistically significant decrease observed four weeks after the intervention was completed. Considering Cohen’s [37] range of effect size, the DHEA-S score of the post-test shows a small-to-medium effect size (ES = 0.255), and the follow-up test shows a larger effect size than the post-test (ES = 0.265).

## 4. Discussion

In this study, we implemented the TSL program to reduce depression and stress in a sample of soldiers and verified its effectiveness. The emotional stability and mitigating effects of oxidative stress achieved through the TSL program were similar to those observed in North Korean female defectors [18] and parents of children with cancer [17]. Although existing studies have primarily focused on family relationships, the present study focused on soldiers’ relationships with peers in the military.

While interventions for relieving soldiers’ depression and stress generally focus on counseling methods for individuals, collaborative tasks in the TSL program (through direct and positive expression) can be useful for soldiers. The TSL program may be an effective intervention because it is based on dynamic interactions between individuals. Owing to the Confucian and hierarchical culture of the Korean military, soldiers generally have difficulty expressing personal problems and finding resources to cope with stress. Consequently, soldiers with high levels of depression and stress are generally recommended individual counseling or risk group-oriented interventions. For example, in the case of soldiers with serious emotional or psychological problems, interventions are conducted in separate camps for approximately four weeks. However, these measures have been criticized as afterthoughts and short-term approaches. Thus, the TSL program is a potential therapeutic and preventive measure to improve soldiers’ mutual relations within the military system.

Overall, the results of this study may indicate that levels of depression are correlated with DHEA-S levels, although the causal directions are not clear from our limited data. In addition, DHEA-S levels were significantly higher after the TSL program. This finding is consistent with previous studies in which emotional variables mediated cognitive processes and directly affected somatization symptoms [38,39,40]. Moreover, the effects of the TSL program included strengthening resilience (by improving the quality of interpersonal relationships) and inducing positive changes in perceptions and attitudes. This can be extremely useful in military settings, where physical and mental abilities are of utmost importance.

The TSL program can be an intervention method to cope with soldiers’ stress, but the Republic of Korea does not have an institution that provides professional counseling services that focus on soldiers’ suicide or maladjustment, nor does it have an expert system that can comprehensively intervene in problems that arise under the characteristics of the military organization, such as emotional problems. Therefore, it is urgently required to introduce a military social worker policy that can apply and link professional counseling services to intervene in psychological difficulties within the military. As a condition for preparing such a system, research on the military welfare system and the establishment of an institution that can cultivate military professional social workers should be preceded.

The United States, which is implementing advanced military social welfare, has established and operated military social welfare centers in universities. The U.S. Military Social Welfare Center provides training for military social workers with financial support from the U.S. Department of Defense, and produces personnel who can conduct military mental health research and professional counseling services. In particular, in the case of UCLA, seven military social welfare-related centers are operated under the university, and the Nathanson Family Resilience Center is conducting community-based military social welfare research and expert training [41]. Therefore, the establishment of specialized institutions for military personnel should be prioritized.

Nevertheless, this study had several limitations. First, owing to the relatively small sample size and the fact that the participants were from the same unit, it is difficult to generalize the results. Thus, future studies should use larger and more representative samples of soldiers from different units. Second, the effectiveness of the program was verified using a two-group randomized experimental design. An alternative might be to use the Randomized Controlled Trial (RCT) study design, which is a trial conducted by randomly dividing the subjects who participated in the clinical trial into an experimental group and several control groups to clearly confirm the effect of the intervention. In RCT, the control group is divided into an active control group and a placebo control group. Future research should include other groups to make an effective comparison. Third, the same individual was the program facilitator and researcher. Therefore, the researcher’s expectations of the participants regarding the observed effects was not controlled, which could have affected the objectivity of the measurements. Future studies should include different researchers to perform each task. Fourth, there may have been a learning effect because the members of the experimental and control groups were colleagues in the same unit. Subsequent research should include other groups from different units to prevent this learning effect from influencing the results. Finally, when the study participants were divided into an experimental group and a control group, random assignments were conducted. However, homogeneity was not secured in the pre-test for depression. These results can be seen as a limitation of the research through surveys. This is also the reason why hormone levels were confirmed through DHEA-S in this study. However, future studies will need to ensure homogeneity in the pre-test. In addition, a self-report questionnaire should be conducted to more elaborately verify the numerical change in stress. Despite these limitations, the TSL program has proven to be an effective approach for improving the adaptability of military soldiers. In particular, it highlights the need to secure support resources (through direct cooperation with team members) and strengthen the emotional competence of individuals.

## 5. Conclusions

In this study, the TSL program strengthened the positive dynamics of the military organization through practical tasks such as encouraging one another and expressing empathy for fellow soldiers. Improving their relationships with their peers gave the participants more opportunities to discover the necessary resources for support during emotional crises. Moreover, realizing the importance of other soldiers’ support and feeling grateful to them played a crucial role in establishing the positive dynamics of military culture. Such aspects allowed the soldiers to appreciate their coexistence and co-prosperity with other soldiers, which had a positive effect on their cognitive ability to cope with stress.

## Figures and Tables

**Figure 1 healthcare-13-00424-f001:**
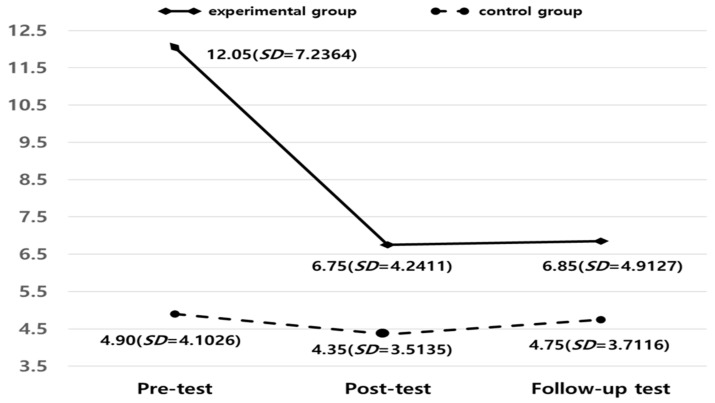
Changes in depression scores. Note: Analysis of covariance (ANCOVA): post-test (*F* (1,39) = 15.459, *p* < 0.01) and follow-up test (*F* (1,39) = 17.694, *p* < 0.01).

**Figure 2 healthcare-13-00424-f002:**
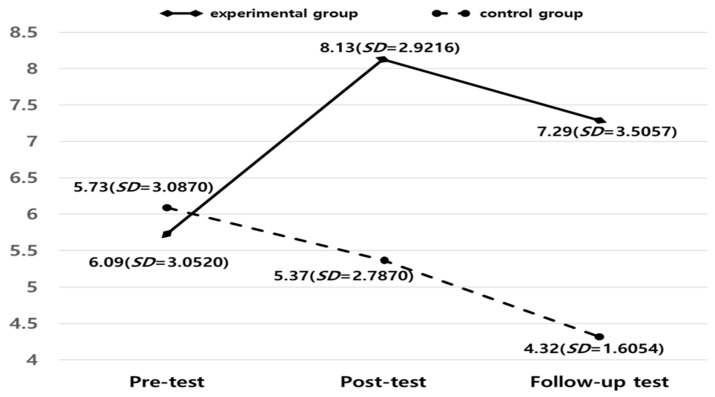
Changes in DHEA-S levels (nm/mL). Note: Analysis of covariance (ANCOVA): post-test (*F* (1,39) = 12.666, *p* < 0.01) and follow-up test (*F* (1,39) = 13.319, *p* < 0.01).

**Table 1 healthcare-13-00424-t001:** Research model.

Group	Participants	Measurement	Pre	Intervention	Post	Follow-Up
Experimental group	20	SQ *	SQ1	O	SQ2	SQ3
		BM **	BM1	O	BM2	BM3
Control group	20	SQ	SQ4		SQ5	SQ6
		BM	BM4		BM5	BM6

* SQ: Self-reported questionnaires (Depression); ** BM: Biomarker (DHEA-S).

**Table 2 healthcare-13-00424-t002:** Sample size calculation of the F-test for ANCOVA.

Test Family	Statistical Test		
F-tests	ANCOVA: Fixed effects, main effects, and interactions		
**Type of power analysis**			
A priori: Compute the required sample size—given α, power, and effect size			
**Input parameters**		**Output parameters**	
Effect size F	0.774	Noncentrality parameters	17.97228
α err probe	0.05	Critical F	3.3541308
Power (1-β err probe)	0.95	Denominator df	27
Numerator df	2	Total sample size	30
Number of groups	2	Actual power	0.9569790
Number of covariates	1		

**Table 3 healthcare-13-00424-t003:** Demographics of the participants.

Variable		Experimental Group *M* (*SD*)/*n* (%)	Control Group *M* (*SD*)/*n* (%)	Range
Military service	Army	10 (50%)	10 (50%)	
	Air force	10 (50%)	10 (50%)	
Age of participants (years)		21.40 (1.96)	20.60 (0.88)	19–28
Rank of participants (levels) *	Private	2 (10.00)	6 (30.00)	-
	Private first class	10 (50.00)	6 (30.00)	-
	Corporal	6 (30.00)	2 (10.00)	-
	Sergeant	2 (10.00)	6 (30.00)	-

* Korea’s military service includes four levels: private, private first class, corporal, and sergeant.

## Data Availability

The data presented in this study are available on request from the corresponding author.

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
