# Peer review of "The Role of Interdisciplinary Convergence for Mental Health Among Korean Military Servicemen: Focusing on Depression and Salivary Dehydroepiandrosterone-Sulfate"

_healthcare, 2025, doi:10.3390/healthcare13040424_

Round 1

Reviewer 1 Report

Comments and Suggestions for Authors

This study explored the effectiveness of the “TSL” intervention for depression and stress among a military sample. Though the research gap was clearly addressed, I still have several concerns. My comments are as below.

1.      The term DHEAs should be clearly explained in the introduction together with its relationship between stress. DHEAs might be a indicator of many variables.

2.      In the abstract, the authors mentioned “improve family relationships”, which seems to be irrelevant to the research aim.

3.      After reading the introduction, readers might believe the dependent variables are depression and stress. Why stress was not measured in the questionnaire survey? It is a pity that this study did not test stress and compare its results with DHEAs.

4.      Please provide more details of sample size calculation in G power. How was the effect size of 0.75 was selected from your citations 17 and 18. What were the other parameters set in G power for the sample size of ANCOVA?

5.      The depression levels were significantly different between the two groups. The experimental group had a clear higher depression level in the pre test (see Figure 1), though no p value was provided. Please clarify the potential impact of such difference in depression on your results.

6.      The significance of the difference in DHEAs in the pre test was also unknown: 5.73 and 6.09. The experimental group and control group might be already different before the intervention.

7.      Although the discussion part addressed some implications, the significance of the present study is still limited. The design of RCT is recommended for such research questions.

8.      There are some grammar errors. Please carefully check about the language again. For example, in lines 11 to 13, “ …to improve family relationships on the psychological and physical health…”. This sentence was not clearly written.

Comments on the Quality of English Language

The English language quality needs to be checked. 

Author Response

Comments 1: The term DHEAs should be clearly explained in the introduction together with its relationship between stress. DHEAs might be a indicator of many variables.

Response 1: We appreciate your thoughtful comments on our manuscript. TAs you mentioned, there was a lack of explanation about the relationship between DHEA-s and stress. DHEA-s is a representative hormone that can check the physical response of stress, and in this study, DHEA-s was used as a measurement variable to compensate for the limitations of the method of checking stress through a self-report questionnaire. Based on the opinions you mentioned, we supplemented them as follows.

“Moreover, in this study, the effectiveness of the experiment was verified through DHEA-s, a representative 'steroid hormone' that responds to body stress. Because their dehydroepiandrosterone-sulfate (DHEA-S) levels could be measured to assess their stress levels and complement the limitations of the self-reported questionnaires. The hormone DHEA-S are primarily utilized as indicators of stress and aging, and problems such as chronic stress and depression occur when the amount remaining in the body decreases [20,21]. The DHEA-S is inhibited by the hypothalamic–pituitary–adrenal axis in response to stress.”

Comments 2: In the abstract, the authors mentioned “improve family relationships”, which seems to be irrelevant to the research aim.

Response 2: As mentioned in the introduction part, the TSL program used in this experimental study focuses on improving family relationship and intervenes. In order not to misunderstand, the abstract has been revised.

Comments 3: After reading the introduction, readers might believe the dependent variables are depression and stress. Why stress was not measured in the questionnaire survey? It is a pity that this study did not test stress and compare its results with DHEAs.

Response 3: We agree with you. Although biological changes in the body were confirmed through the DHEA-s hormone to compensate for the limitations of the self-report questionnaire, the questionnaire is also needed to confirm the stress changes more precisely. The importance of having to measure stress changes through the questionnaire was added to the limitations of the study.

“In addition, a self-report questionnaire should also be conducted to more precisely verify the numerical change of stress.”

Comments 4: Please provide more details of sample size calculation in G power. How was the effect size of 0.75 was selected from your citations 17 and 18. What were the other parameters set in G power for the sample size of ANCOVA?

Response 4: As you advised, I added a detailed explanation for the sample size calculation. In addition, we made it possible to visually check the G power calculation through the table. The effect size was calculated through the effect size calculation formula for F-test(ANCOVA). Number of covariate is one, and no other variables were added.

Comments 5: The depression levels were significantly different between the two groups. The experimental group had a clear higher depression level in the pre test (see Figure 1), though no p value was provided. Please clarify the potential impact of such difference in depression on your results.

Response 5: As emphasized throughout this study, it is our point in this study that self-report questionnaires such as depression have limitations. However, in your opinion, the failure to secure homogeneity of depression variables is a clear limitation of the study. The potential impact of differences in pre-test of depression on outcomes are specified as follows.

“This means that homogeneity between groups was not secured(t=-3.844, P<0.01).Therefore, there is a limitation in analyzing changes between groups of depression in post- and follow-up tests. However,”

Comments 6: The significance of the difference in DHEAs in the pre test was also unknown: 5.73 and 6.09. The experimental group and control group might be already different before the intervention.

Response 6: For DHEA-S, homogeneity was secured in pre-test. We added pre-test P value of DHEA-S.

Comments 7: Although the discussion part addressed some implications, the significance of the present study is still limited. The design of RCT is recommended for such research questions.

Response 7: Thank you for your advice on the RCT design that we didn't consider. We added the contents of the RCT design to the limitations of the study.

Comments 8: There are some grammar errors. Please carefully check about the language again. For example, in lines 11 to 13, “ …to improve family relationships on the psychological and physical health…”. This sentence was not clearly written.

Response 8: We made grammar errors, which are the most basic parts. Overall, we reviewed the sentences.

Reviewer 2 Report

Comments and Suggestions for Authors

Thank you for submitting your manuscript: Interdisciplinary Convergence for Mental and Family Health among Korean Military Servicemen: Focusing on Depression and Salivary DHEAs. The manuscript is important mental health research among military personnel with two intervention and control groups, which is very interesting. I thoroughly enjoyed reading your manuscript. I have the following feedback to improve your manuscript before publishing in this journal.

Abstract section:

·        The methods section should be clear and robust because this is an experimental study, which is not clear except for the sample size, but clearly come up with research design, sampling approaches, duration of your interventions, and outcome analysis.

·        The findings section is very limited, yet it should be clearly stated with significant values instead of a qualitative tone.

·        Provide a pragmatic recommendation of your research that has not yet been covered.

 Introduction section:

·        The first two paragraphs of the introduction section are adequately articulated; however, the next paragraph (lines 64-75) should clearly describe the reason for this study rather than any methodological specifics, which you can explain in the methods section.

Methods section:

·        This is one of the intervention-based studies, so clearly state and adhere to the research methods and any guidance. You can start with 2.1 research design, study participants/sites, sample size, and sampling strategy (with a flow chart—treatment/intervention and control group), intervention details, data collection—tools, data management, and analysis—which makes the section more coherent and covers it appropriately.

·        Move table 1 to the finding section, which could not be in the research design section, and make the research design section more clear: Which experimental research have you designed and implemented using credible sources to back up?

·        Clearly describe the data collection process with biological markers, and how did you collect that information?

Results section:

·        Should start with demographic characteristics of respondents and move to study outcomes. Starting with descriptive statistics makes it helpful to follow your analytical part rather than only analytical findings.

 Discussion section:

·        Looks well attempted, but the discussion would be stronger if you looked at global literature around military (security personnel) mental health burdens around the world, compared and contrasted, and provided your thoughts to make the discussion section much more clear.

 Conclusion section:

The conclusion section is missing in this manuscript. So I would suggest making a succinct conclusion and providing pragmatic recommendations to address the burden that makes your manuscript much clearer.

Author Response

Comments 1: [Abstract section] The methods section should be clear and robust because this is an experimental study, which is not clear except for the sample size, but clearly come up with research design, sampling approaches, duration of your interventions, and outcome analysis.

Response 1: We appreciate your thoughtful comments on our manuscript. As you mentioned, the contents of the abstract were insufficient. I added information about research design, etc.

Comments 2: [Abstract section] The findings section is very limited, yet it should be clearly stated with significant values instead of a qualitative tone.

Response 2: We dded important significant values to the Results section.

Comments 3: [Abstract section] Provide a pragmatic recommendation of your research that has not yet been covered.

Response 3: As you advised, we have added clinical guidelines to the Korean military social work, which is still in its early stages.

Comments 4: [Introduction section] The first two paragraphs of the introduction section are adequately articulated; however, the next paragraph (lines 64-75) should clearly describe the reason for this study rather than any methodological specifics, which you can explain in the methods section.

Response 1: In the introduction, the basic contents of the program were described because it was necessary to explain the counseling technique used in this study. However, as you mentioned, the purpose or necessity of the study was insufficient, so we added it.

Comments 5: [Methods] This is one of the intervention-based studies, so clearly state and adhere to the research methods and any guidance. You can start with 2.1 research design, study participants/sites, sample size, and sampling strategy (with a flow chart—treatment/intervention and control group), intervention details, data collection—tools, data management, and analysis—which makes the section more coherent and covers it appropriately. Move table 1 to the finding section, which could not be in the research design section, and make the research design section more clear: Which experimental research have you designed and implemented using credible sources to back up?, Clearly describe the data collection process with biological markers, and how did you collect that information?

Response 5: As you advised, I moved Table 1 to the research results. To supplement the explanation of the experimental research, the Materials and Methods section has been supplemented.

Comments 6: [Results section] Should start with demographic characteristics of respondents and move to study outcomes. Starting with descriptive statistics makes it helpful to follow your analytical part rather than only analytical findings.

Response 6: I agree with you, so I have revised the order of the text so that we can start with demographic characteristics.

Comments 7: [Discussion & Conclusion section] Looks well attempted, but the discussion would be stronger if you looked at global literature around military (security personnel) mental health burdens around the world, compared and contrasted, and provided your thoughts to make the discussion section much more clear. The conclusion section is missing in this manuscript. So I would suggest making a succinct conclusion and providing pragmatic recommendations to address the burden that makes your manuscript much clearer.

Response 7: For the conclusion of the study, we added an example of the United States leading military social work to the Discussion part.

Round 2

Reviewer 1 Report

Comments and Suggestions for Authors

Thanks. The revised version reads much better. 

Author Response

Thank you for being in charge of the article review and giving me good advice.